# A 3D-Printed Integrated Handheld Biosensor for the Detection of *Vibrio parahaemolyticus*

**DOI:** 10.3390/foods13111775

**Published:** 2024-06-05

**Authors:** Yuancong Xu, Qian Zhang, Yunyi Li, Xiaoxu Pang, Nan Cheng

**Affiliations:** 1College of Chemistry and Life Science, Beijing University of Technology, Beijing 100124, China; xuyuancong@bjut.edu.cn; 2Key Laboratory of Safety Assessment of Genetically Modified Organism (Food Safety), Ministry of Agriculture, Beijing 100083, China; bestwishestoyou123@163.com; 3Beijing Laboratory for Food Quality and Safety, College of Food Science and Nutritional Engineering, China Agricultural University, Beijing 100083, China; liyunyi9905@163.com (Y.L.); pxxyaojiayou@163.com (X.P.)

**Keywords:** 3D-printed biosensor, RPA-CRISPR/Cas12a, lateral flow strip, *Vibrio parahaemolyticus*, rapid detection

## Abstract

*Vibrio parahaemolyticus* (*V. parahaemolyticus*) is one of the important seafood-borne pathogens that cause a serious gastrointestinal disorder in humans. Recently, biosensors have attracted serious attention for precisely detecting and tracking risk factors in foods. However, a major consideration when fabricating biosensors is to match the low cost of portable devices to broaden its application. In this study, a 3D-printed integrated handheld biosensor (IHB) that combines RPA-CRISPR/Cas12a, a lateral flow strip (LFS), and a handheld device was developed for the ultrasensitive detection of *V. parahaemolyticus*. Using the preamplification of RPA on *tlh* gene of *V. parahaemolyticus*, a specific duplex DNA product was obtained to activate the trans-cleavage activity of CRISPR/Cas12a, which was then utilized to cleave the ssDNA probe. The ssDNA probe was then detected by the LFS, which was negatively correlated with the content of amplified RPA products of the *tlh* gene. The IHB showed high selectivity and excellent sensitivity for *V. parahaemolyticus* detection, and the limit of detection was 4.9 CFU/mL. The IHB also demonstrated great promise for the screening of *V. parahaemolyticus* in samples and had the potential to be applied to the rapid screening of other pathogen risks for seafood and marine environmental safety.

## 1. Introduction

*Vibrio parahaemolyticus* (*V. parahaemolyticus*), a Gram-negative halophilic bacterium that widely exists in coastal and marine waters. It is also one of the important seafood-borne pathogens that cause a serious gastrointestinal disorder in humans [1,2]. Due to the high levels of marine consumption over coastal areas, *V. parahaemolyticus* has been transmitted through contaminated marine foods [3]. Numerous reports have shown that *V. parahaemolyticus* has caused many foodborne disease outbreaks which account for over half of all bacterial food poisoning outbreaks and has even been the leading cause of travelers’ diarrhea in many countries [4,5]. Thus, there is a great requirement for the precise detection and tracing of *V. parahaemolyticus* to ensure the safety of coastal environments and marine foods.

Recently, the detection strategies of *V. parahaemolyticus* have transitioned from traditional culture-based methods to non-culture-based methods, such as immunological methods, spectroscopy methods, nucleic acids amplification techniques, and biosensor technologies [6,7,8,9,10,11]. Given the importance of rapid analysis to marine food supply chain and coastal environment monitoring, it is necessary to ensure that the detection methods have higher specificity, flexibility, and practicality. Culture-based methods are the gold standard for detecting bacteria, which are time-consuming, labor-intensive and less sensitive. Spectroscopy is a convenient tool for bioanalytical applications and is particularly good at single-sample analysis, where it can be used to separate individual microsamples from mixed samples by spectral characterization [12]. Biosensor-based technologies have obvious performance advantages such as accuracy, sensitivity, and point-of-care testing and have become the main technology in *V. parahaemolyticus* detection [13]. However, there are still some unsatisfactory shortcomings of this methods, such as complex operations and specialized equipment dependence. Thus, providing an integrated biosensor that can be operated without special training or instruments is critical to contribute to the detection of *V. parahaemolyticus*.

The clustered regularly interspaced short palindromic repeats (CRISPR) and CRISPR-associated protein (Cas) system (CRISPR/Cas) is an adaptive immune system that has been widely found in most bacteria and archaea [14]. Due to its superior targetability, the CRISPR/Cas system has been widely used in gene editing, medical diagnostics, and biosensing analysis [15,16,17]. Among all CRISPR/Cas systems, CRISPR/Cas12a is most commonly used in the development of biosensors due to its ability to target DNA characteristics. Cas12a protein (Cpf1) guided by a single-stranded guide CRISPR RNA (crRNA), which specifically binds with the target double-strand DNA (dsDNA) with the PAM site (5′-TTTN) or the single-strand DNA (ssDNA), exhibits cis-cleavage activity to cleave target DNA and trans-cleavage activity to indiscriminately cleave any nontarget ssDNA [18].

Biosensors are usually made up of a biological element with a physiochemical transducer. Combining biological signal recognition and amplification with physiochemical signal transformation and output is preferable for simplifying operations in the reaction system. Isothermal amplification methods, which remove the dependency on expensive thermal cycling instruments, have been widely used in nucleic acid amplification [19,20]. Through rational design, amplification products can be the target of the CRISPR/Cas system, thereby realizing the specific nucleic acid information translated into optical signals [21,22,23] and electrochemical signals [24]. Therefore, the blend of isothermal amplification and the CRISPR/Cas system is conducive to developing simple, sensitive, and convenient CRISPR/Cas-based biosensors. Typical applications are SHERLOCK [25] and DETECTR [26] and HOLMESv2 [27]; however, their results heavily depend on the expensive fluorophore-labeled ssDNA, which limits the extension and application of on-site detection. In our previous study, a single-strand nucleic acid-based lateral flow strip (LFS) combined with recombinase polymerase amplification (RPA) for the ultrasensitive detection of food pathogens was introduced [28]. It is a beneficial operation and can realize the visual detection of the unlabeled ssDNA, which decreases the cost significantly. Here, we continued to apply RPA for signal amplification and the LFS for signal output while introducing CRISPR/Cas12a for signal transduction to realize the simple and rapid detection of *V. parahaemolyticus*.

Aside from the requirement of simplicity and stability, another major consideration when fabricating biosensors is to match a low-cost portable device to broaden its application in water, food, and environmental safety control [29]. For CRISPR/Cas-based biosensors, many portable devices have been designed, such as lateral flow devices [30], microfluidic devices [31], smartphone devices [32], nanopore devices [33], and glucose meters [34]. However, all these devices are designed as readout tools for detection results. The nucleic acid extraction process and detection process still have to be completed by relying on laboratory instruments. Some integrated instruments also have been developed, such as reversible valve-assisted chips [23] and Cas12a-PB [35], to realize on-site detection. These instruments simplify nucleic acid detection but still require fluorescence analysis, and cannot be combined with the cheaper LFS. Therefore, there is still plenty of room for improvement regarding integrated CRISPR/Cas-based biosensors.

In this paper, we focused on the design of a handheld 3D-printed device, which can replace the PCR instrument to realize nucleic acid amplification, greatly simplifying the operating equipment and improving its applicability on site. At the same time, it integrates RPA-CRISPR/Cas12a and LFS analysis. On the one hand, it realizes visualization and rapid analysis. On the other hand, it is possible to avoid aerosol contamination by opening the lid during the reaction. Furthermore, an integrated handheld biosensor (IHB) combining RPA-CRISPR/Cas12a, LFS, and a 3D-printed handheld device was developed for the convenient and visual detection of *V. parahaemolyticus* on site. First, a federated approach was established for rapid DNA signal amplification, which could complete RPA in 10 min and CRISPR/Cas12a cleavage in 10 min. Then, the cleavage products were qualitatively detected on the LFS. Last, a handheld 3D-printed device was designed to realize the integrated operation of RPA-CRISPR/Cas12a-LFS. Therefore, this design can perform a simple, sensitive, and rapid detection of *V. parahaemolyticus* and may facilitate significant advancements in water, food, and environmental safety control.

## 2. Materials and Methods

### 2.1. Materials

*V. parahaemolyticus* and other bacterial strains used in this study are listed in Appendix A. The nucleic acid sequences were synthesized by Beijing Tsingke Biotech Co., Ltd. (Beijing, China), and are listed in Appendix A. A Wizard magnetic DNA purification kit was purchased from Promega (Madison, WI, USA). A TwistDx TwistAmpTM Liquid Basic kit was purchased from TwistDX Ltd. (Cambridge, UK). Cas12a (Cpf1) was ordered from ToloBio (Shanghai, China). Other related materials are listed in the Appendix A.

### 2.2. RPA-CRISPR/Cas12a Procedure

Genomic DNA was extracted according to our previous method and device [36,37]. An RPA assay was carried out with the TwistAmpTM Liquid Basic kit in a Bio-Rad S1000TM thermal cycler (BIO-RAD, Richmond, CA, USA) and the handheld 3D-printed device. The RPA master mix was prepared by adding 25 μL of 2× reaction buffer, 9.2 μL of dNTP_S_, 5 μL of 10× Basic E-mix, 2.5 μL of 20× core reaction mix and 2.4 μL of each primer at 10 μM. In the RPA assay, the experiment was performed in a 10 μL solution containing 9.3 μL of RPA master mix, 0.2 µL of the DNA template, and 0.5 μL of magnesium acetate (MgOAc, 280 mM). Since the RPA reaction started as soon as MgOAc was added, the tubes were immediately incubated for 10 min at 37 °C.

CRISPR/Cas12a assay was performed using a Bio-Rad CFX96 (Bio-Rad, Richmond, CA, USA) and the handheld 3D-printed device. The CRISPR master mix was prepared by adding 8 μL of 10× Tolo Buffer, 5 μL of Cas12a (1 μM), 5 μL of crRNA (1 μM), 0.8 μL of cut probe (or LP, 10 μM), and 51.2 μL of ddH_2_O. The trans-cleavage of CRISPR/Cas12a (80 μL) was further performed with 10 μL of RPA products by adding 70 μL of CRISPR master mix. And, the reactions were incubated for 10 min at 37 °C.

### 2.3. Preparation of AuNPs and ssDNA-Modified AuNPs

AuNPs were prepared according to a previously established method [28]. A round-bottom flask was soaked in an acid solution overnight, washed with tap water and ultrapure water, filled with ultrapure water to the neck of the round-bottom flask, and heated until boiling. The solution that contained 190 mL ultrapure water and 2 mL HAuCl_4_ (1 wt%) was heated to boiling. Then, 8 mL trisodium citrate solution (1 wt%) was added to heat for 10–15 min, then the mixture was cooled down at room temperature. When the solution turned cherry red, it was aliquoted and stored at 4 °C for later use.

ssDNA-modified AuNPs were prepared according to Liu’s method with modification [38]. The thiolated ssDNA are listed in Appendix A (AP). In a typical experiment, 100 μL of the prepared AuNPs was mixed with 20 µL of AP (100 µM). The mixture was then placed in a laboratory freezer (set at −80 °C) for 15 min, followed by thawing at room temperature. Then, it was centrifuged at 10,000× *g* for 5 min and we discarded the supernatant. At last, the precipitate was dissolved in 45 µL of resuspended solution (20 nM Na_3_PO_4_ + 5% BSA + 0.25% Tween-20 + 10% sucrose). The AP-modified AuNPs (AuNP-AP) were stored at 4 °C.

### 2.4. Lateral Flow Strip Procedure

For the LFS, 2.5 μL of AuNP-AP was dripped onto the conjugate pad in advance. Then, 120 μL of running buffer containing 4× SSC, 1% BSA, and 0.05% Tween 20 with 40 µL of the RPA-CRISPR/Cas12a product solution was applied to the sample pad. During the assay process, the solution with the AuNP-AP complexes migrated through the test line and control line in sequence by capillary force. AuNPs were captured on the test line when the LP hybridized to complementary sequences within AP and TCP. Therefore, the presence of the LP in the products was determined with a visual red band. The color depth of the red bands on the test strip was quantified with ImageJ 1.53 software.

### 2.5. Design of the 3D-Printed Handheld Device

A 3D-printed handheld device was designed to realize the integrated operation of RPA-CRISPR/Cas12a-LFS. The 3D-printed handheld device included three parts, “Spiral grass”, “Soil base” and “Water source”. The “Spiral grass” consisted of five essential ingredients: a spiral scaffold, tricolor capillary tubes, a steel pin latch, an LFS, and a sample tube. In part of the tricolor capillary tubes, the yellow tube contained 9.3 μL of the RPA master mix, the red tube contained 70 μL of the CRISPR master mix, and the black tube contained 120 μL of the running buffer.

### 2.6. Application of the Integrated Biosensor in V. parahaemolyticus-Spiked Oyster Samples

To ensure the practicability of the proposed integrated biosensor, *V. parahaemolyticus*-spiked oyster samples were analyzed. Oysters were purchased from the local supermarket in Beijing, China, and then autoclaved for 20 min at 121 °C to ensure they were microorganism-free. Then, the sterilized oysters were washed with sterile water three times. *V. parahaemolyticus* strain was added into the LB broth and incubated in a constant-velocity shaker at 37 °C for 10 h. Then, the cultured solution was diluted 10-fold with saline, and the concentration of the *V. parahaemolyticus* was determined by the conventional plate counting method. For the real sample test, three levels of *V. parahaemolyticus* between 10^2^ and 10^4^ CFU/mL were spiked into the oyster samples for assays. A total of 25 g oyster and 25 mL of *V. parahaemolyticus* were aseptically transferred into a sterile filter bag and homogenized in 200 mL of sterile PBS buffer for 2 min using a stomacher. Subsequently, DNA was extracted from 1 mL of the filtrate and tested following the above-described biosensor procedure.

First, water was injected into the water tank and heated to 37 °C by the heating electrode, and this temperature was maintained long enough for further reaction. For the RPA reaction, 0.5 μL of DNA was added to the sample tube, and RPA solution was forced from the purple spiral capillary. Then, the tube was placed into water to start amplification. After 10 min, CRISPR/Cas12a solution was forced from the yellow spiral capillary and incubated for 10 min. Next, the running buffer was forced from the pink spiral capillary, and the bolt was removed, making the LFS fall into the tube. Due to capillary force, the mixture in the sample tube migrated upward along the LFS. After 3 min, the red bands of the test line and the control line were observed by the naked eye. The color depth of the test line was analyzed quantitatively by ImageJ software. Three parallel tests and a negative control were performed for each sample, and the average value of the three tests was calculated.

## 3. Results and Discussion

### 3.1. Mechanism of the 3D-Printed Integrated Handheld Biosensor (IHB)

In this report, we described the construction of a 3D-printed integrated handheld biosensor (IHB) that combined the amplification features of RPA-CRISPR/Cas12a with visual detection on an LFS and was completed in a handheld device. The design of this innovative integration and the device are shown in Figure 1.

As shown in Figure 1A, the 3D-printed handheld device included three parts. (1) “Spiral grass”: This consisted of five essential ingredients: a spiral scaffold, a tricolor capillary, a steel pin latch, an LFS, and a sample tube. The yellow capillary, red capillary, and black capillary, respectively, are equipped with the RPA master mix, the CRISPR master mix, and the running buffer. The mouths of the three capillaries were closed by Vaseline. The sample tube was inserted into the bottom of the spiral scaffold, the LFS was placed into the cavity of the spiral scaffold, and the steel pin latch was sheathed into the spiral scaffold to keep out the LFS. (2) “Soil base”: This was a base stent that had grooves in the four corners. The “spiral grass” and “soil base” were connected through a tongue-and-groove connection. (3) “Water source”: This was a water tank with temperature control that could heat water with a portable power source. A dropper could be used to force the relevant reagent into the sample tube at a specified time in practice. The operation realized the consistency of the experiment without opening the cover of the sample tube, avoiding aerosol pollution. The diagram with all the dimensions labeled, the real object, and the actual video of this 3D-printed handheld device are provided in the Appendix A.

After preprocessing (Figure 1B), the *V. parahaemolyticus* DNA was analyzed using RPA- Cas12a/crRNA-LFS detection. The principle and design of the detection are described in Figure 1C. DNA and MgOAc solution were added to the sample tube first. Then, the RPA reagent, the CRISPR/Cas12a reagent, and the running buffer were added into the tube sequentially. Finally, the LFS was released into the sample tube. During the RPA-CRISPR/Cas12a process, *V. parahaemolyticus* DNA was first amplified by RPA. To ensure the specificity of the detection, a specially designed crRNA was introduced to target the RPA products. In the presence of RPA products, they would complement the crRNA to form the Cas12a/crRNA-DNA complex, exhibiting trans-cleavage activity to cleave the ssDNA probe (LP) indiscriminately. As a result, AuNP-AP only complemented CCP on the control line (CL), displaying a red band of AuNPs. In contrast, the intact LP complemented AP and formed the LP/AuNP-AP complex on the conjugate pad. Then, the complex was paired with TCP on the test line (TL) to form the TCP/LP/AuNP-AP complex, causing the test TL to display the first red band. The excess AuNP-AP continued to migrate and be captured on the CL, forming a second red band. In this way, the results could be easily distinguished via the red band of TL differences between positive and negative samples.

### 3.2. Development of RPA-CRISPR/Cas12a

It is very important to choose suitable primers during nucleic acid amplification. Following the RPA guidelines, three sets of primers (Appendix A) were designed, and all were applied to select suitable primers for the next cleavage reaction. The results of the RPA were analyzed by 2% agarose gel electrophoresis and are shown in Figure 2A. All the primers obtained positive bands whose lengths were expected. The comparison of the band intensity in RPA products declared that the VP-F1/R1 primers had the highest amplification efficiency. Therefore, we next verified the specificity of the VP-F1/R1 primers (Figure 2B) and confirmed that they were ideal primers.

Then, the feasibility of RPA-CRISPR/Cas12a was ensured, utilizing the fluorescence of the cut probe (FAM and BHQ-labeled LP). The fluorescence was continuously collected by a Bio-Rad CFX96 (Bio-Rad, Richmond, CA, USA) every 30 s for 1 h, and the fluorescence intensity at 518 nm was then quickly measured using a Thermo Scientific Varioskan Flash (Thermo Scientific, Waltham, MA, USA) with excitation at 492 nm. Under 365 nm UV light irradiation, the severed probe emitted green fluorescence (Appendix A). Then, a series of photos were taken with a smartphone (iPhone 12) under 365 nm UV light in a dark room. As shown in Figure 2C,D, in the presence of *V. parahaemolyticus*, the fluorescence intensity increased gradually and ultimately reached a saturated value (red line). The results suggested that Cas12a was activated by the RPA products of *V. parahaemolyticus* and showed strong trans-cleavage activity, leading to an obvious fluorescence signal. However, in the absence of *V. parahaemolyticus*, the fluorescence response was undetectable because the fluorescence of the intact cut probe was highly quenched (blue line). Furthermore, the visual analysis was consistent with the above phenomenon, with no fluorescence at the start of the cleavage reaction, and the emissions increased after 1 h, except for the negative control.

### 3.3. Optimization of RPA-CRISPR/Cas12a

To obtain the proper detection system and reaction time, it is necessary to optimize the RPA-CRISPR/Cas12a system, and the best detection parameters are shown in Figure 3. Generally, the RPA assay gave the best results at 37 °C with an extension time of 20 min. Considering the time cost, we first explored the reaction time of RPA. As shown in Figure 3Ba,C, the fluorescence intensity was increased with the RPA time. There were significant differences in the fluorescence intensity for the 10 min and 5 min (**, *p* < 0.01) but no differences for the 10 min and 15 min (ns, *p* > 0.05). When the RPA time was extended to 20 min, again, there was an obvious difference compared to the 10 min (*, *p* < 0.05). Considering the time cost, the most appropriate RPA reaction time was 10 min. The Cas12a-crRNA complex determines the number of RuvC active cutting sites and plays a crucial role in the performance of the trans-cleavage. In published studies, there are several choices for the ratio of Cas12a to crRNA. Then, we further investigated the ratio of Cas12a to crRNA, ranging from 1:1 to 1:3. As shown in Figure 3Bb,D, the ratio of Cas12a to crRNA showed a trivial impact on the increasing rate of the trans-cleavage. There were no significant differences in the fluorescence intensity for the different ratios, indicating that they had the same amount of the Cas12a/crRNA complex. Since the complex was combined by equal parts of Cas12a and crRNA, Cas12a:crRNA was selected as 1:1 in subsequent studies.

The Cas12a-crRNA complex determines the number of RuvC active cutting sites and the effectiveness of trans-cleavage. The fluorescence intensity was associated with differences in trans-cleavage effectiveness. Then, we attempted to analyze the trans-cleavage ability with Cas12a-crRNA complex concentrations ranging from 12.5 nM to 75 nM. As shown in Figure 3Bc,E, the fluorescence intensity increased with increasing Cas12a/crRNA concentration. However, no significant difference in fluorescence intensity was observed between 62.5 nM and 75.0 nM (ns, *p* > 0.05). Furthermore, according to the fluorescence curve (Appendix A), when the concentration of the Cas12a-crRNA complex was lower than 50 nM, the reaction proceeded slowly and did not reach a plateau in 40 min. When the concentration of the Cas12a-crRNA complex was higher than 50 nM, the reaction could obtain a maximum fluorescence intensity in 10 min. Therefore, considering the effectiveness of the CRISPR/Cas system, the 62.5 nM Cas12a-crRNA complex was chosen for the subsequent studies. Then, the trans-cleavage performance of the 62.5 nM Cas12a-crRNA complex was further evaluated with the cut probe concentration ranging from 100 nM to 600 nM. As shown in Figure 3Bd,F, the fluorescence intensity was proportional to the cut probe concentration, and the results were consistent with the kinetics of an enzyme-catalyzed reaction. However, the appropriate concentration still needs further optimization when combined with the LFS.

### 3.4. Characterization of the AuNPs and AuNP-AP

AuNPs were prepared by the citrate reduction method. The prepared AuNP solution is cherry red in color (Appendix A). The particle sizes and morphology of the prepared AuNPs were characterized by transmission electron microscopy (TEM) at 200 kV. As shown in Figure 4A, the prepared AuNPs were well dispersed and had no aggregation. The TEM image was analyzed by ImageJ, and the average diameter of the AuNPs was determined to be approximately 14 nm. Subsequently, the UV–visible absorption spectra of AuNPs and AuNP-AP were measured by UV–vis spectroscopy (Pgeneral, Beijing, China). As seen in Figure 4B, the maximum absorption wavelength of the AuNPs was 523 nm. After conjugation, AP covered the whole surface of the AuNPs, and the maximum absorption wavelength of AuNP-AP was 531 nm. The redshifted maximum absorption wavelength demonstrated the successful conjugation of AP to AuNPs. At the same time, the AuNP concentration was determined by UV–vis absorbance via the Beer–Lambert law (*A = εlc*), where *A* is the AuNP maximum absorbance at λ_max_, *ε* is the extinction coefficient, and *l* is the path length. The ε value for AuNPs with a diameter of 14 nm was calculated to be 2.95 × 10^8^ M^−1^ cm^−1^ by a previous study [39], and similar AuNPs (diameter = 13.4 ± 1.6 nm) were 2.01 ×10^8^ M^−1^ cm^−1^ [40]. Then, the AuNP concentration was estimated to be 9.45 nM.

### 3.5. Optimization of the Lateral Flow Strip

To accommodate the LFS, the unlabeled ssDNA probe was used as the LP to complement AP and form the LP/AuNP-AP complex on the test line. Once the LP was cleaved, only one red band could be obtained on the control line (Figure 5A). Therefore, it was important to improve the analytical performance of the LFS system, including the concentration of LP, the type of running buffer, the amounts of CRISPR/Cas12a products, and the amount of AuNP-AP.

Firstly, the LP concentration was validated again by comparing the negative group and the positive group. Theoretically, when the LP was cleaved by Cas12a, the TL of the negative group had a red band, but the TL of the positive group had no band. As shown in Appendix A, the LP concentration was optimized under different CRISPR cut times, and the redness of positive group TL lightened with increasing the CRISPR cut time. When the concentration of LP was set to 600 nM, 300 nM, and 100 nM, there was no band on the positive group TL with the cut time for 50 min, 30 min, and 10 min, respectively. This result indicated that the LP of the positive group was cleaved completely and unable to form the TCP/LP/AuNP-AP complex on the TL. Concerning the time cost of the rapid detection, 10 min should be acceptable for our method. Under this condition, the concentration of the LP was adjusted from 25 nM to 500 nM for further analysis (Figure 5Ba,C). The TL color depth was analyzed by ImageJ software, and the peak area was used as a quantitative index. In the negative experiment, the redness of TL deepened with increasing LP concentration. However, the color of TL also turned red when the LP concentration was higher than 200 nM in the positive control group. This result indicated that the LP was not cleaved completely and formed the TCP/LP/AuNP-AP complex on the TL. As a result, the concentration of the LP should be lower than 200 nM to match the theoretical analysis. Comparing the shades of red on the negative TL, 100 nM TP was chosen as the optimal concentration, which also improved the sensitivity.

Since there was no red band on the positive group TL, only the negative group TL was analyzed in subsequent experiments. Then, seven buffer types were compared, including 4× SSC (1), 4× SSC + 1% trehalose (2), 4× SSC + 1% BSA (3), 4× SSC + 0.05% Tween 20 (4), 4× SSC + 1% BSA + 0.05% Tween 20 (5), 4× SSC + 1% BSA + 0.05% Tween 20 + 0.002% Triton x-100 (6), and 4× SSC + 1% BSA + 0.05% Tween 20 + 0.002% Triton x-100 + 10 mM Tris-HAc + 5 mM Kac (7). As shown in Figure 5Bb,D, only four buffer types revealed an obvious red band on the TL. According to the peak areas of the TL, the best running buffer was buffer d. However, the best running buffer was 4× SSC + 1% BSA + 0.05% Tween 20 after comparison with four CLs. As shown in Figure 5Bc,E, the resuspended solution amount of AuNP-AP from 30 μL to 120 μL was optimized, and there was a correlation between the peak area and AuNP-AP. With the increase in the volume of resuspended solution, the concentration of AuNP-AP decreased. When the amount of resuspended solution was more than 45 μL, the red color of the TL gradually lightened. When the amount of resuspended solution was less than 45 μL, the TL had an unmistakable red band. Therefore, the optimal amount of resuspended solution was 45 μL. As shown in Figure 5Bd,F, when the amount of CRISPR/Cas12a product added was less than 10 μL, there was only a limited number of LPs to pair with TCP, which led to the low signal intensity on the TL. When more than 20 μL of the products were added, the peak area of the TL reached saturation. Considering that the volume of the RPA-CRISPR/Cas12a system in the sample tube of the 3D-printed handheld device was 80 μL, to avoid sample transfer and provide ease of operation, the optimal amount of 80 μL of product was selected for subsequent LFS detection. This was also the reason why we injected 120 μL of running buffer into the pink spiral capillary.

### 3.6. Sensitivity of the IHB for V. parahaemolyticus Detection

In the optimal conditions mentioned above, the sensitivity of the IHB was tested with 10-fold serial dilutions of *V. parahaemolyticus* samples (ranging from 10 to 1.0 × 10^9^ CFU/mL). The red color and relative peak area of the TL in response to different *V. parahaemolyticus* concentrations are shown in Figure 6A,B. The relative peak area is PN-PP, where PN is the peak area in the negative control and PP is the peak area in the presence of *V. parahaemolyticus*. With the increase in the concentration of *V. parahaemolyticus*, the red color of the TL gradually lightened, and the relative peak area increased. The logarithmic scales showed that the relative peak area exhibited a linear correlation (R^2^ = 0.9930) when the concentration of *V. parahaemolyticus* increased from 1.0 × 10^1^ to 1.0 × 10^6^ CFU/mL. The regression equation was expressed as Y = 1634X + 1136, where Y denotes the relative peak area, and X signifies the logarithm of the *V. parahaemolyticus* concentration. Thus, the limit detection limit (3σ/slope, where σ represents the standard deviation of the blank) of the presented biosensor was 4.9 CFU/mL. Appendix A lists the various detection methods [7,8,10,11,21,41,42,43,44,45] of *V. parahaemolyticus* assays that have been reported, including limit of detection (LOD), detection time, and suitability for on-site testing. As shown in Appendix A, the LOD of this IHB is as low as 4.9 CFU/mL, which is much lower than many reported methods, and is sufficient to meet the food safety testing requirements.

### 3.7. Specificity of the IHB for V. parahaemolyticus Detection

Then, the specificity of the IHB was further verified under the optimal experimental conditions. The experiments were performed to detect other pathogenic bacteria, including *Salmonella* spp., *Listeria monocytogenes*, *Pseudomonas aeruginosa*, *Escherichia coli*, *Staphyloccocus aureus*, *Shigella sonnei*, *Clostridium perfringens*, *Bacillus cereus,* and *Enterobacter sakazakii* at a concentration of 1.0 × 10^5^ CFU/mL. As shown in Figure 6C,D, for all the samples, both the TL and CL exhibited red bands, except for *V. parahaemolyticus*, in which only the CL exhibited a red band. Therefore, the IHB can effectively distinguish *V. parahaemolyticus* from other pathogenic bacteria, confirming the specificity of the proposed biosensor.

### 3.8. Real Sample Analysis by the IHB

A test using the presented IHB was performed on oyster samples to investigate its practicability. Fresh oyster samples were processed following the steps mentioned earlier. As shown in Table 1, the results demonstrated that *V. parahaemolyticus* in oysters could be accurately detected. The IHB for detecting *V. parahaemolyticus* achieved recoveries between 99% and 106%, with RSDs between 1.93% and 4.35%. Therefore, the developed biosensor demonstrated a high ability to detect *V. parahaemolyticus* in real foods.

## 4. Conclusions

In summary, an IHB strategy was developed for conveniently and visually detecting *V. parahaemolyticus*. The proposed strategy combines the advantages of RPA-CRISPR/Cas12a-LFS and a 3D-printed handheld device. The features of this IHB include rapid nucleic acid amplification, visible biosensor analysis, and a compact detection device. One of the greatest advantages of this integrated design is the ability to avoid aerosol contamination caused by opening the lid during the reaction process. Moreover, the 3D-printed device is cheap and reusable. However, one inconvenience in the whole design is that the injection of reaction reagents is cumbersome. The operation of the method could be further simplified if the reaction reagent could be made into a lyophilized powder pre-sealed in the capillary. The *tlh* gene was employed as a recognition element in the biosensor, achieving high specificity. The RPA-CRISPR/Cas12a approach was established for signal amplification and transduction. A colorimetric signal readout was achieved in the LFS by the difference in AuNP accumulation. Moreover, the operation of RPA-CRISPR/Cas12a-LFS could be completed in the 3D-printed handheld device, achieving the simple, rapid, and inexpensive on-site detection of *V. parahaemolyticus*. After optimizing the experimental parameters, the IHB was able to detect *V. parahaemolyticus* more accurately and specifically. The LOD reached a level as low as 4.9 CFU/mL of *V. parahaemolyticus*. Furthermore, with the aid of a 3D-printed device, the detection time of IHB can be reduced to 23 min. The analysis of artificially contaminated oysters proves that this IHB is well suited for the on-site detection of *V. parahaemolyticus*. All the results of this study indicate the excellent reliability and practicability of the IHB. The proposed IHB method is versatile and can be extended to the detection of other foodborne pathogens by substituting reaction reagents. And, this biosensor may also provide a practical model for the future examination of nucleic acid risk affecting the analysis of food and the environment.

## Figures and Tables

**Figure 1 foods-13-01775-f001:**
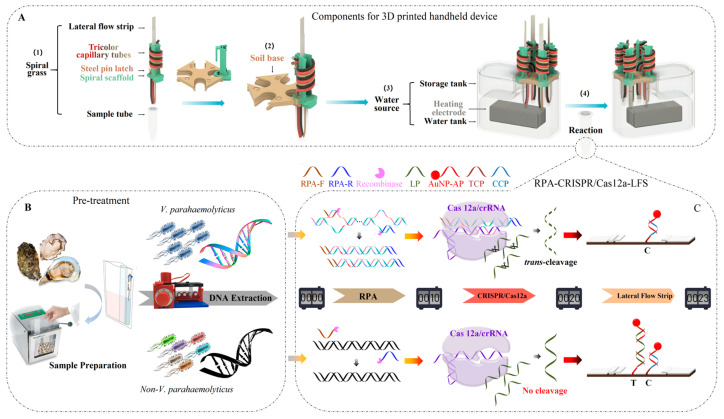
Schematic diagram of *Vibrio parahaemolyticus* detection with the integrated handheld biosensor. (**A**) Design of the 3D-printed handheld device. (**B**) Sample processing. (**C**) The detection principle of the integrated handheld biosensor.

**Figure 2 foods-13-01775-f002:**
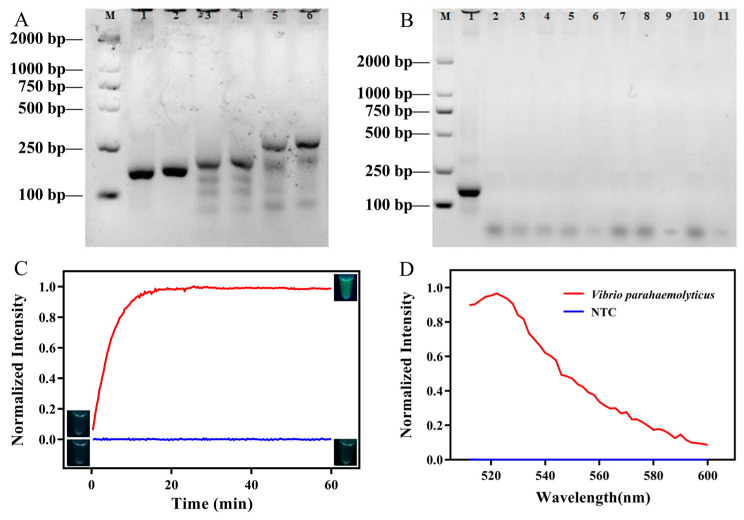
(**A**) Primer screening in the RPA. Lane M: DL2000 DNA maker; Lanes 1 and 2: VP-F1/VP-R1 primers; Lanes 3 and 4: VP-F2/VP-R2 primers; Lanes 5 and 6: VP-F3/VP-R3 primers; (**B**) specificity verification of VP-F1/VP-R1 primers. Lanes 1–11: *Vibrio parahaemolyticus*, *Salmonella* spp., *Listeria monocytogenes*, *Pseudomonas aeruginosa*, *Escherichia coli*, *Staphyloccocus aureus*, *Shigella sonnei*, *Clostridium perfringens*, *Bacillus cereus*, *Enterobacter sakazakiidd* and ddH_2_O; (**C**) time monitoring curve of fluorescence change; (**D**) fluorescence spectrum with or without *Vibrio parahaemolyticus*.

**Figure 3 foods-13-01775-f003:**
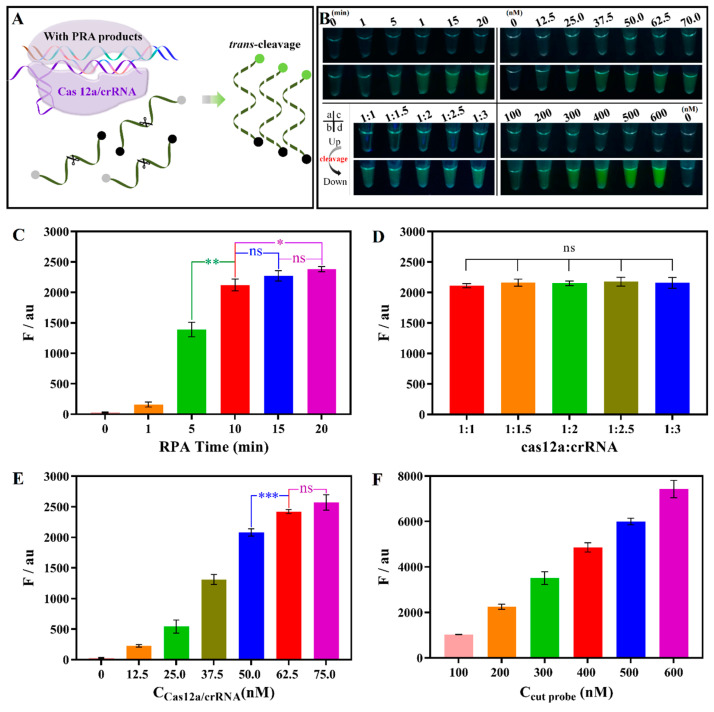
Optimization of the RPA-CRISPR/Cas12a cleaving system (original reaction system: Cas12a/crRNA 50 nM, cut probe 200 nM, cut time 40 min). (**A**) Scheme of the trans-cleavage; (**B**) photographs before and after trans-cleavage with excitation at 365 nm. (**a**) RPA time; (**b**) Cas12a:crRNA; concentrations of the Cas12a/crRNA (**c**) and cut probe (**d**); (**C**) fluorescence intensity of different RPA time; (**D**) fluorescence intensity of different Cas12a:crRNA ratios; (**E**) fluorescence intensity of different Cas12a/crRNA concentrations; (**F**) fluorescence intensity of different cut probe concentration. (Data were analyzed using two-tailed test, “***” means *p* < 0.001, “**” means *p* < 0.01, “*” means *p* < 0.05, “ns” means *p* > 0.05).

**Figure 4 foods-13-01775-f004:**
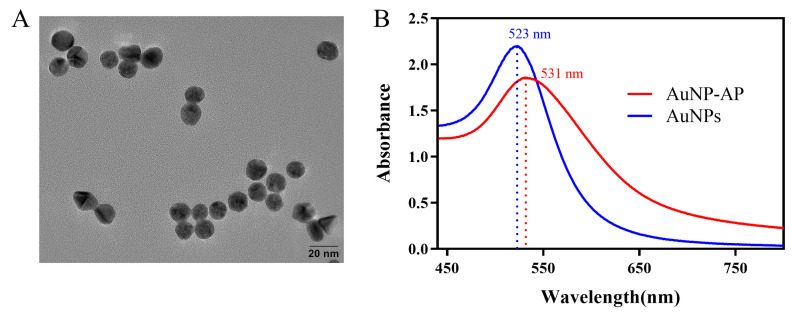
Characterization of AuNPs and AuNP-AP. (**A**) TEM images of AuNPs; (**B**) absorption spectrum of AuNPs and AuNP-AP.

**Figure 5 foods-13-01775-f005:**
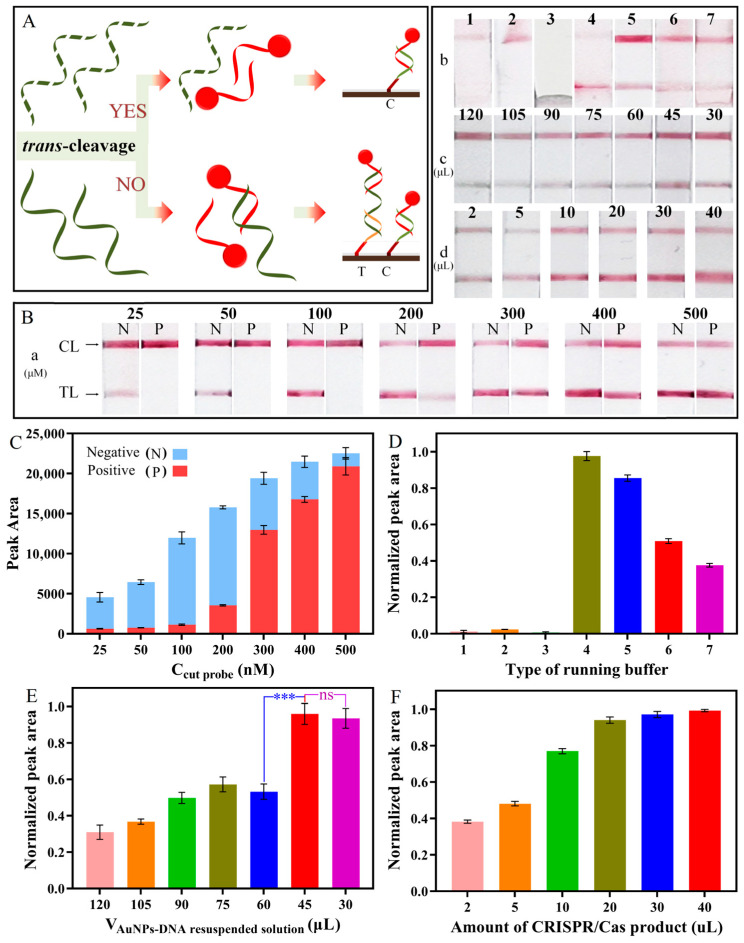
Optimization of the RPA-CRISPR/Cas12a-based LFS. (**A**) Scheme of the CRISPR/Cas12a-based LFS; (**B**) photographs of LFS. (**a**) Concentrations of the LP; (**b**) running buffer; (**c**) the volume of AuNP-DNA resuspended solution; (**d**) the amount of CRISPR/Cas product; (**C**) peak area of TL with different concentrations of the LP; normalized peak area of TL with different (**D**) running buffer, 1: 4× SSC, 2: 4× SSC + 1% trehalose, 3: 4× SSC + 1% BSA, 4: 4× SSC + 0.05% Tween 20, 5: 4× SSC + 1% BSA + 0.05% Tween 20, 6: 4× SSC + 1% BSA + 0.05% Tween 20 + 0.002% Triton x-100, and 7: 4× SSC + 1% BSA + 0.05% Tween 20 + 0.002% Triton x-100 + 10 mM Tris-HAc + 5 mM Kac. (**E**) volume of AuNP-DNA resuspended solution, and (**F**) amount of CRISPR/Cas product. (N stands for negative group and P for positive group. Data were analyzed using two-tailed test, “***” means *p* < 0.001, “ns” means *p* > 0.05).

**Figure 6 foods-13-01775-f006:**
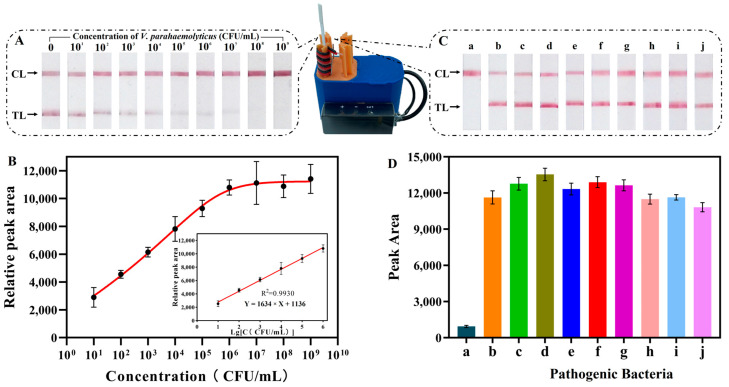
Sensitivity and specificity of the IHB in response to *V. parahaemolyticus*. (**A**) Photographs of lateral flow strip and (**B**) relative peak area of TL with different concentration of *V. parahaemolyticus*; (**C**) photographs of LFS and (**D**) peak area of TL with different pathogenic bacteria. (**a**) *V. parahaemolyticus* (1.0 × 10^9^ CFU/mL); (**b**–**j**) *Salmonella* spp., *Listeria monocytogenes*, *Pseudomonas aeruginosa*, *Escherichia coli*, *Staphyloccocus aureus*, *Shigella sonnei*, *Clostridium perfringens*, *Bacillus cereus,* and *Enterobacter sakazakii* (1.0 × 10^5^ CFU/mL).

**Table 1 foods-13-01775-t001:** Detection results of oyster samples.

Samples	Spiked (CFU/mL)	Measured (CFU/mL)	Recovery of the Assay (%)	RSD (%)
Oyster sample 1	0	ND *		
	10^2^	106.03 ± 2.05	106	1.93
Oyster sample 2	0	ND *		
	10^3^	1027.12 ± 31.91	103	3.11
Oyster sample 3	0	ND *		
	10^4^	9937.49 ± 432.14	99	4.35

* ND indicates no results.

## Data Availability

The original contributions presented in the study are included in the article/Appendix A, and further inquiries can be directed to the corresponding author.

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
