# Peer review of "A 3D-Printed Integrated Handheld Biosensor for the Detection of Vibrio parahaemolyticus"

_foods, 2024, doi:10.3390/foods13111775_

Round 1

Reviewer 1 Report

Comments and Suggestions for Authors

The paper presents a 3D printed biosensor for detecting Vibrio parahaemolyticus. Detailed results of the detection process are presented. The analysis appears to be convincing. However, since several papers in the literature have discussed the detection of Vibrio parahaemolyticus, the novelty of the work does not appear significant. The time and device cost appears to be the main selling point of the current study. However, it does not appear that all the required analysis can be performed by the device alone. I would recommend the authors to clarify this in the manuscript and highlight the novelty and contribution of the work in the introduction section more clearly. In addition to this, I also suggest a few other minor revisions listed below:

1. The schematic diagram of the 3D printed device is missing in the main text. Please provide a diagram with all the dimensions labelled.

2. The resolutions of Scheme 1 and Fig. 1 are very poor. Please upload much higher resolution images. Also, all figures should be captioned as figures (and not scheme). Hence, scheme 1 should be labelled as Fig. 1 and all subsequent figures should be labelled accordingly.

3. Would it be possible to provide some images related to AuNP preparation? The “cherry red color” of the solution would be better expressed with a photograph. In addition, the TEM images of the nanoparticles should be moved to the main text. Also, comment on the concentration of the AuNPs in the solution.

4. Please comment what kind of base-pair resolution is achievable using agarose gel electrophoresis. How does this compare to polyacrylamide gel electrophoresis (PAGE)?

5. Please mention the model number of the digital camera used to record the fluorescence. The excitation wavelength is stated to be 365 nm for the camera. What was the emission wavelength that was recorded by the camera?

6. Is it possible to provide a cost estimate of the device and operation (including all the reagents, 3D printing etc.)? How does this compare with other methods? Is the device reusable?

7. Vibrio parahaemolyticus detection using various techniques have been reported in the literature. For example, surface plasmon sensors, spectroscopy methods (including single sample spectroscopy) etc. have enhanced detection capabilities. Some of these techniques should be cited in the paper. For example:

i. https://doi.org/10.1021/ac3021888

ii. https://doi.org/10.1166/jnn.2018.14212

iii. https://doi.org/10.1063/5.0191871

Comments on the Quality of English Language

English appears fine

Author Response

Thank you very much.

Reviewer 2 Report

Comments and Suggestions for Authors

Dear Authors,

Manuscript foods-2978439, with the title 'A 3D-printed Integrated Handheld Biosensor for the Detection of Vibrio parahaemolyticus' presents a 3D-printed integrated handheld biosensor (IHB) that was developed for ultrasensitive detection of Vibrio parahaemolyticus, a seafood-borne pathogen causing gastrointestinal disorders. The IHB combines RPA-CRISPR/Cas12a, a lateral flow strip, and a handheld device. The IHB showed high selectivity and excellent sensitivity, with a limit of detection of 4.9 CFU/mL. It has potential for rapid screening of other pathogen risks for seafood and marine environmental safety. The paper presents an integrated handheld biosensor (IHB) for the rapid detection of V. parahaemolyticus on site. It combines RPA-CRISPR/Cas12a, a lateral flow strip, and a 3D-printed device, enabling simple, sensitive, and rapid detection, potentially advancing water, food, and environmental safety control.

The authors developed an integrated biosensor (IHB) strategy for visually detecting V. parahaemolyticus using RPA-CRISPR/Cas12a, a lateral flow strip, and a 3D-printed handheld device. This strategy combines rapid nucleic acid amplification, visible biosensor analysis, and a compact detection device. The tlh gene was used for high specificity, and the RPA-CRISPR/Cas12a approach was used for signal amplification and transduction. The ssDNA probe was designed as a ligation probe, capturing gold nanoparticles (AuNPs) on the lateral flow strip. The 3D-printed handheld device completed the operation, achieving simple, rapid, and inexpensive on-site detection of V. parahaemolyticus.

The manuscript is nicely arranged, and in addition to having great illustrations, it also has a textual description that is very clear. Both the graphics and the figures are of a quality that is commensurate with the journal's standards, and the references provide support for the statements that are capitalized.
After conducting a thorough technical review of the material, I would like to suggest that these findings be taken into consideration for FOODS magazine.

Upon completion of the text's technical review, I will accept.

Author Response

Thank you for your recognition of our study.

Reviewer 3 Report

Comments and Suggestions for Authors

Dear Authors,

The main objective of this work was to use an integrated handheld biosensor (IHB) combining RPA-96 CRISPR/Cas12a, a lateral flow strip and a 3D-printed handheld for visual detection of V. parahaemolyticus.

In my opinion, this manuscript is well written with an original or relevant topic within the field.

This article contributes an innovative method for detecting Vibrio parahaemolyticus and in my point of view can be published.

I would like to state just some information that could be better presented in the text of the manuscript, such as:

Page 5:

Schematic diagram 1 is not readable and needs to be rewritten.

Page 11:

Lines 400-403: This part of the text refers to the methodology and not results.

Conclusions need to be rewritten according to the results obtained.

Author Response

Thank you very much.

Reviewer 4 Report

Comments and Suggestions for Authors

The manuscript reports on an original research work to develop an integrated handheld biosensor (IHB) combining RPA-96 CRISPR/Cas12a, a lateral flow strip and a 3D-printed handheld device for on-site detection of V. parahaemolyticus. The authors propose a design that enables simple, sensitive, and rapid detection of V. parahaemolyticus and can be used in a variety of ways to improve water, food, and environmental safety control.

The introduction is adequate to provide the proper context for the research, and the objectives are clearly stated. The Materials and Methods section describes the main experimental trials and provides all the necessary details. The research is sound, and the results are presented in detail and are appropriate for the objectives sought. Overall, the discussion of the results could be improved and better supported with literature references, including comparisons with other authors' results in terms of sensitivity, detection time and ease of use, limit of detection and limit of quantification. The conclusions could be improved by discussing in more detail the strengths and weaknesses of the proposed biosensor design and their implications for other areas. The manuscript contains some typographical errors that can be corrected by a complete revision.        

Overall, the work is solid, and the results obtained are an important contribution. The manuscript could be improved to increase its impact.

Author Response

Thank you very much.

Round 2

Reviewer 1 Report

Comments and Suggestions for Authors

The paper has been improved. Most of the recommended changes have been incorporated. However, the authors missed one of the recommended references in their citation list. Although that specific work does not focus on   vibrio parahaemolyticus, it discusses single sample spectroscopy which is relevant to the subject matter.

Comments on the Quality of English Language

English is fine

Author Response

Thank you very much. I'm very sorry for our mistake. We have cited the reference and added the relevant content.

Line 48-50: Spectroscopy is a convenient tool for bioanalytical applications, and is particularly good at single-sample analysis, where it can be used to separate individual microsamples from mixed samples by spectral characterization[12].

[12] Zaman, M. A., Wu, M., Ren, W., Jensen, M. A., Davis, R. W., Hesselink, L. Spectral tweezers: Single sample spectroscopy using optoelectronic tweezers. Appl. Phys. Lett. 2024, 124, 071104.
